# Physical Activity Levels and Psychological Well-Being during COVID-19 Lockdown among University Students and Employees

**DOI:** 10.3390/ijerph191811234

**Published:** 2022-09-07

**Authors:** Adrián De la Rosa, Armando Monterrosa Quintero, María Alejandra Camacho-Villa, Coralie Arc-Chagnaud, André Gustavo Pereira de Andrade, Sergio Reyes-Correa, Ronald Quintero-Bernal, Juan Pedro Fuentes-García

**Affiliations:** 1Laboratory of Exercise Physiology, Sports Science and Innovation Research Group (GICED), Unidades Tecnológicas de Santander (UTS), Bucaramanga 680006, Santander, Colombia; 2Research Group SER-SICIDE, Department of Physical Education and Sports, Universidad Católica de Oriente (UCO), Rionegro 054040, Antioquia, Colombia; 3Laboratoire MOVE, Faculté des Sciences du Sport, Université de Poitiers, F-86000 Poitiers, France; 4Biomechanics Laboratory, Universidade Federal de Minas Gerais, Belo Horizonte 31270-901, Brazil; 5Physical Activity and Sport Program, Sports Science and Innovation Research Group (GICED), Unidades Tecnológicas de Santander (UTS), Bucaramanga 680006, Santander, Colombia; 6Didactic and Behavioral Analysis of Sports Research Group (ADICODE), Faculty of Sport Sciences, University of Extremadura, 10003 Cáceres, Spain

**Keywords:** anxiety, stress, mental health, comfort, public health

## Abstract

During the lockdown for the coronavirus disease 2019 (COVID-19), entire populations were instructed to live in home confinement. We investigated the effects of the COVID-19 lockdown on the physical activity (PA) and mental health of students and employees in a Colombian University. A cross-sectional study was conducted through an online survey during the first isolation. A total of 431 respondents (192 males) aged 18–60 years old (28.1 ± 11.1 years) participated. The international Physical Activity Questionnaire (IPAQ) and the short version of the Psychological General Well-Being Index (PGWBI-S) were used. The lockdown had a negative effect on PA levels, with students exhibiting the greatest decrease (~34%; *p* ˂ 0.001) compared to employees (~24%; *p* ˂ 0.01). The analysis showed a greater change in PA behavior before and during the lockdown in highly active student participants (5750 vs. 5141 MET min/week; *p* < 0.05). Additionally, the psychological assessment revealed a lower score in students compared to employees in the male (70.1 vs. 82.6) and female groups (60.2 vs. 79.6). Moreover, the results revealed an influence of sex, with only the female students exhibiting a state of distress. Self-reported PA and psychological well-being were compromised during the COVID-19 lockdown in the academic community, with students and females being more affected.

## 1. Introduction

Coronavirus 2019 (COVID-19) is an infectious disease caused by a novel form of coronavirus [1]. Since its onset in China in late 2019, the worldwide outbreak of the COVID-19 pandemic has had a great impact on people’s physical behaviors and psychological well-being [2]. One of the main strategies adopted by governments to control the spread of the disease was implementing measures such as lockdowns, which proved to be highly effective [3].

During the lockdown, the world’s population was advised to keep social distance and stay home for about 3 to 4 months, except for essential trips for buying food, going to the pharmacist, or to the hospital. Only key workers who performed essential tasks for society were allowed to go to their workplaces. People were required to work and study from home, children could not go to school, and their parents had to combine remote working with home-schooling [4]. Thus, the lockdowns greatly impacted all areas of society, from education and sports to general community life [5,6]. These extreme measures led to changes in lifestyle, which resulted in poor mental health, and reduced physical activity (PA) of the population worldwide [7,8].

For example, reports from [9] indicated increases of 33.7% and 28.4% in the prevalence of major depressive disorder and anxiety disorders, respectively, in the United States, due to the COVID-19 pandemic between 2020 and 2021. Likewise, after evaluating a Saudi university’s academic community during the COVID-19 lockdown, [10] found symptoms of anxiety, depression, and insomnia of 58.1%, 50.2%, and 32.2%, respectively. Additionally, a reduction in PA levels was also reported during the lockdown of students and university employees [11].

Before this pandemic, physical inactivity was already described as a global public health problem [12,13]. In Colombia, according to the National Nutrition Survey [14], approximately half of the population aged between 18 and 64 years old were physically inactive, according to World Health Organization (WHO) recommendations. In addition, both physical inactivity and lockdown have been reported to have had a negative impact on mood in the general population [15], while PA and long-term exercise were shown to have had a positive impact on both physical health and psychological well-being [16,17,18,19].

In Colombia, the first case of COVID-19 was confirmed on 6 March 2020, while the health emergency was declared on 12 March 2020. The increase in positive COVID-19 cases resulted in stricter confinement policies that forced all Colombian citizens to be quarantined between 11 March and 1 September 2020 [20]. In this way, Colombia was one of the countries with the longest lockdown period in Latin America during the pandemic.

The Colombian government allowed individual physical activities only at home for the entire population, while people aged between 18 and 69 years old were allowed to be physically active outdoors only from 1 September 2020 onwards. Therefore, it can be expected that such severe restrictions may have had a negative influence on PA levels and mental health in the general population.

Moreover, Colombian universities were closed, and both university students and employees were sent home to continue studying and working remotely. In this way, people had to adapt to new learning and work environments, which have been shown to generate stress and anxiety [21]. In 2020, global research on the quality of digital well-being measured the digital quality in 85 countries in terms of infrastructure, connectivity, speed, quality, affordability, and security; this same research ranked Colombia as one of the worst countries in this area (#62) [22]. In this sense, learning and work efficiency would decrease among members of the academic communities, and this could cause states of anxiety, stress, or depression.

Despite the aforementioned, no studies have reported the impact of the COVID-19 lockdown on PA levels and psychological well-being in a sample from a Colombian University’s Academic Community. Therefore, the present study was thus aimed to identify the changes in PA levels during the lockdown and investigate the prevalence of self-reported mental health conditions in a sample of Colombian university students and employees during the lockdown.

## 2. Materials and Methods

### 2.1. Study Design

This was an observational and analytical cross-sectional research study, where an electronic survey was sent to the isolated academic community via email for completion. A non-probability sampling method was used

### 2.2. Procedures

The participants were recruited by distributing an invitation through administrative channels of the Universidad Católica de Oriente through social media (WhatsApp, classroom, and institutional email). A standardized questionnaire was created within Office 365^®^ (Microsoft Office professional plus 2019), including validated tools to assess PA (i.e., the *International Physical Activity Questionnaire–Short Form* [IPAQ-SF], which contains 7 questions and designed to assess changes in PA) and mental well-being (i.e., the *Psychological General Well-Being Index-Short version* [PGWBI-S]) [23].

The online survey was sent to potential respondents (employees and students) between 8 April 2020 and 15 April 2020, and it was accessible via an anonymous link distributed via email. The survey was announced by the Human Resources (HR) department and physical education faculty and was open for a period of 14 days (from 8 April to 22 April 2020) during the COVID-19 emergency in Colombia. Automated reminders were sent two times during this period of time. A brief introduction section prior to the different questionnaires explained the objective of the survey.

Survey data collection was halted on 26 April 2020, and the data were downloaded. Any participant that was missing data for one or more of the items of interest (i.e., physical activity, sedentary behavior, psychological well-being, age, bodyweight, height, etc.) in the survey was subsequently removed from the dataset.

In addition, to participate, the participants had to meet the following inclusion criteria: (i) being a member of the academic university community; (ii) being between the age of 18–65 years old, healthy (physically, psychologically, and cognitively) and to live in Rionegro-Antioquia; (iii) not having been infected with COVID-19; and (iv) being confined for at least one month at the time of data collection. All the participants were previously informed about the purpose of the study and the benefits associated with the research by signing an electronic informed consent form. The study was conducted in accordance with the declaration of Helsinki, and the protocol was approved by the Ethics Committee for Human Beings of the Universidad Católica de Oriente through Resolution 8430/1993 of the Ministry of Health and Social Protection of Colombia. The online data collection was performed using the same questions in the form of Microsoft Forms Office365^®^ (Microsoft Office Professional Plus 2019 for NA and Google Forms^®^ (Google LLC, Mountain View, CA, USA).

Participants were divided into three age groups: young < 21 (n = 83); young adults aged 21–40 years old (n = 275), and adults aged 41–60 years old (n = 75) [8]. In the employee groups, no young adult participants were included, as they were not part of the university’s employee population (Table 1).

### 2.3. Physical Activity Levels

The PA levels were assessed using the Physical Activity Questionnaire (IPAQ) Short Form, which contained 7 questions and was designed to assess changes in PA, previously outlined as valid in this context [24,25]. The questions allowed the assessment of PA levels by providing information on minutes per day or days per week, at any time of the day, devoted to activity before and during lockdown. The participants reported the frequency and duration of different types of activity: vigorous (i.e., heavy lifting, intense aerobic exercise, cycling or treadmill use); moderate (i.e., carrying light loads and cycling at a regular pace, gardening); walking activities; as well as the average time spent sitting on a weekday, including sitting at work [8]. The IPAQ-SF algorithm was used to transform the continuous data into categorical data (i.e., low, moderate, and high PA) [26]. The outcomes were calculated as the weekly metabolic equivalent of a task in minutes (MET min/week). Based on IPAQ recommendations for the scoring protocol, the participants were classified according to PA level: low (<600 MET min/week), moderate (>600 MET min/week), and high (>3000 MET min/week) [27].

### 2.4. Psychological General Well-Being Index (PGWBI-S)

The short version of the Psychological General Well-Being Index (PGWBI-S) [23] was used to assess the self-perceived psychological well-being of both students and university employees during lockdown. Briefly, this version consists of six dimensions of the original PGWBI test and a subset of six items, which are scored on a six-point Likert scale, ranging from 0 to 5, for a maximum of 30 points. The total score was obtained from the sum of the scores of the six dimensions (anxiety, vitality, depression, self-control, well-being, and general health) by multiplying the overall score by 3.66 (to make it comparable to the original version of the PGWBI). Subsequently, the participants were rated based on a six-level scale: values below 60 indicated strong distress; values between 60 and 69 indicated distress; values between 70 and 89 indicated a state of no distress; values of 90 and above indicated a state of positive well-being [23].

In the PGWBI-S, the six items correspond to item numbers 5, 6, 7, 18, 20, and 21 of the original scale and are the following:

I.Have you been bothered by nervousness or by your ‘‘nerves’’ during the past month?II.How much energy, pep, or vitality did you have or feel during the past month?III.I felt downhearted and blue during the past month.IV.I was emotionally stable and sure of myself during the past month.V.I felt cheerful, lighthearted during the past month.VI.I felt tired, worn out, used up, or exhausted during the past month.

The internal consistency of the PGWB-S test applied to the population in our study was examined with Cronbach’s alpha (α), which should be at least 0.7 to indicate the satisfactory homogeneity of the items within the total scale [28]. In our study, Cronbach’s alpha for reliability was 0.87, indicating good reliability and content validity [29]. A critical value for significance of *p* < 0.05 with a 95% confidence interval (95% CI) was considered significant.

### 2.5. Statistical Analysis

Data analysis was performed using SPSS software (SPSS 25/IBM, Chicago, IL, USA). For each of the variables, the values are shown as mean ± standard deviation or n (%). The normality of distribution was verified with the Kolmogorov–Smirnov test, and homogeneity of variance was tested with Levene’s statistics. The interaction of gender and age (between-subjects factor) in the changes in PA levels before and during the lockdown (within-subjects factor) were analyzed by a mixed design ANOVA (gender × age × time). A multivariate analysis of variance (MANOVA) test was performed to evaluate the PGWBI score and all six dimensions found in the PGWBI-S test.

## 3. Results

The potential participants were n = 518 (Figure 1). However, after excluding 87 datasets because of missing values for physical activity, psychological well-being, age, bodyweight, height, or sex, the final sample was n = 431 (students n = 298 (69.1%) and employees n = 133 (30.9%), mean age 23.7 ± 6.5 years old and 39.9 ± 11.0 years old; mean height 168.1 ± 25.2 cm and 166.3 ± 8.2 cm; mean weight 64.7 ± 12.9 kg and 69.4 ± 12.9 kg; and BMI 23.1 ± 3.7 and 25.0 ± 3.7, respectively), which underwent detailed analysis.

### 3.1. Physical Activity Was Modified by the Lockdown: Are Sex, Age, or Role Influential?

The examination in the entire sample showed statistically significant differences in total PA for sex (*p* < 0.0001), time (*p* < 0.0001) and for the time × sex interaction (*p* < 0.0001). The post hoc analysis revealed significant intra-group before-during differences in total PA in both students (3441 vs. 2347 MET min/wk; *p* < 0.001) and employee groups (2389 vs. 1834 MET min/wk; *p* < 0.01). Likewise, the post hoc analysis revealed significant inter-group differences between students and employees (*p* < 0.001), as shown in Figure 2A.

Regarding the PA levels (low, moderate, and high), the statistical analysis showed significant differences between before and during lockdown only for highly active participants in the student group (5750 vs. 5141 MET min/wk; *p* < 0.05), as shown in Figure 2B.

When the sex factor was included, the analysis showed that total PA significantly decreased during the lockdown in both male and female students and employees, as shown in Figure 2C. Overall, the mixed ANOVA analysis revealed an effect for time (*p* < 0.001) and also for the time × role interaction (*p* < 0.05), but no for the time × sex (*p* > 0.05) or time × sex × role interactions (*p* > 0.05). Multiple pairwise comparisons showed significant intra-group differences before vs. during the lockdown in both male and female student groups (*p* < 0.001 in both cases) and male and female employee groups (*p* < 0.05 in both cases). As shown in Figure 2D, the lockdown modified total PA levels in both male and female students according to age category. The ANOVA test showed an effect for the time category (*p* < 0.001) but no for the time × sex, time × age category or time × sex × age category. Multiple comparisons revealed significant intragroup differences before-during the lockdown in total PA for the young and young adult male and female groups (*p* < 0.001 in both cases) and in the adult female group (*p* < 0.05).

In the employee groups, the ANOVA revealed an effect only for the time factor (*p* < 0.01). Multiple pairwise comparisons showed significant intra-group differences only for the young male adult group (*p* < 0.01), as shown in Figure 2D.

### 3.2. Psychological Well-Being Is Strongly Influenced by Role and Sex

The scores of the six health domains, comprising anxiety, vitality, depression, self-control, well-being, and general health, and the PGWBI score of the participants, divided by gender and role, are shown in Figure 3.

Overall, psychological well-being affected students more than employees during the lockdown. The MANOVA procedure showed an effect for the role in the PGWBI score (*p* < 0.001) and anxiety (*p* < 0.001), vitality (*p* < 0.001), depression (*p* < 0.001) self-control (*p* < 0.001), positive well-being (*p* < 0.001), and general health dimensions (*p* < 0.001). In the student groups, the MANOVA test revealed significant differences in the PGWBI score (*p* < 0.001) and anxiety (*p* < 0.001), vitality (*p* < 0.05), self-control (*p* < 0.001), positive well-being (*p* < 0.05), and general health dimensions (*p* < 0.01), with the female group being more affected. However, in the employee groups, the statistical analysis showed no differences in any dimension or PGWBI score between the male and female groups.

According to the role variable, the MANOVA procedure revealed statistical inter-group differences (*p* < 0.05 and *p* < 0.01) between employee and student groups in most dimensions and PGWBI score with the student groups exhibiting the lowest results.

The results according to age category and PA levels (low, moderate, and high) during lockdown are shown in Table 2 and Table 3, respectively. For weight and BMI covariates, no additional changes were found.

Briefly, according to age category in both male and female student and employee groups, the MANOVA procedure showed no statistical differences between groups (Young, young adult, and adult) in the PGWBI score or any dimension (*p* > 0.05), as shown in Table 2. However, according to the sex variable, the analysis revealed differences in the PGWBI score and all the six dimensions for young, young-adult, and/or adult groups, only in the student groups (*p* < 0.05 and *p* < 0.01).

Finally, statistical differences (*p* < 0.05 and *p* < 0.01) were found according to role (students vs. employees) in the PGWBI score and all the six dimensions in male and/or female, young-adult, and adult groups, with both young adult and adult employee groups exhibiting better mental well-being in terms of the PGWBI score, as compared to student groups.

On the other hand, with respect to the PA level during the lockdown, the MANOVA test showed statistical intra-group differences (low, moderate, and high) in the male and female student groups in the PGWBI score and some dimensions (*p* < 0.05 and *p* < 0.01). as for the employee groups, only statistical differences were found in the vitality dimension for the female groups (*p* < 0.01), as shown in Table 3.

## 4. Discussion

This study aimed to assess the changes in PA levels and psychological well-being during the lockdown related to the COVID-19 pandemic among students and university employees. As expected, the findings of this study are consistent with several previous studies that have reported changes in PA levels and psychological health issues due to the lockdown.

### 4.1. Physical Activity and Lockdown

Previous studies showed the negative impact of the lockdown on PA. In this regard, Ref. [8] informed that Italian people reduced their total PA, as well as their moderate and high PA levels during the lockdown, while a Spanish study reported that university students reduced their moderate (−29.5%) and vigorous (−18.3%) PA during lockdown [2]. In the same way, a systematic review reported a decrease in PA levels during the lockdown in university students from Australia, Croatia, England, Hungary, Italy, Mexico, Spain, and the USA [30]. However, increases in PA levels were reported in Spanish university students and Belgian adults during lockdown [31,32], while no changes in PA levels in Swiss office workers were reported [33].

The results of the present study show that PA levels were negatively impacted by the lockdown. Furthermore, they are in line with most of the previous findings mentioned above.

Among the main findings, we found that the lockdown affected total PA levels in the entire sample (Figure 2A), as previously reported in the literature in children and adolescents [34], young adults [35], and university students and employees [36]. However, similar research failed to find changes in PA levels in both university students and employees [11,33].

Overall, our results indicated a reduction in total PA in both students and employees, with students being more affected: ~32% (from 3441 to 2347 MET min/week) vs. ~24% (from 2389 to 1824 MET min/week), respectively. The greater change in PA behavior between before and during lockdown was observed in highly active student participants (n = 140 vs. n = 97, no data shown), as shown by the total MET min/week (5750 vs. 5141; *p* < 0.05) (Figure 2B). However, despite a generally consistent reduction in PA in this study, highly active participants in the employee group succeeded in achieving similar statistical PA levels during the lockdown. The number of subjects reporting a high PA during lockdown decreased as compared to before lockdown (47 vs. 32, no data shown) in the employee group; but curiously, the total MET min/week was maintained (4472 vs. 4654; no statistical differences were found). Several possible explanations arise in this regard. Firstly, this finding could be supported by the fact that it is possible that the employee group would have maintained the amount of high-intensity exercise during the lockdown in order to compensate for the loss of their freedom to move around.

Secondly, it is also feasible that factors such as working/home-schooling from home had a stronger and positive influence on older-people age groups. Recently, a longitudinal study with 5395 participants aged 41 years old in the United Kingdom reported that older individuals were significantly more likely to maintain and then increase their PA levels during the lockdown [37].

In our study, the mean age of the employee participants was higher than the students (39.7 vs. 23.7, no data shown). Therefore, age is a factor that must be important when analyzing the results. In order to fill this knowledge gap, the participants in our study were grouped according to age range, in addition to sex. Our results showed that the adult student group was less affected as compared to the young and young adult groups (Figure 2D). In the same way, male adults in the employee group achieved similar statistical levels of PA during lockdown than before, unlike the young adult group (Figure 2E). In this sense, this picture suggests that policy interventions should be better focused on the long-term impact of restrictions on younger groups, as previously reported [38].

With regards to the sex factor, the lockdown affected both male and female participants in the student and employee groups, with females being more affected (Figure 2D). In this respect, our results are consistent with other reports [8,39], although they are a discrepancy with previous studies [2]. Perhaps men and women in our study had different motivations, and the environment had a stronger influence on one gender over the other. In this sense, there is some evidence on motives for PA according to gender, which showed that some variables, such as challenge, competition, social recognition, strength, and endurance, motivated men but not women, while weight management was the main motivation for women [40]. Nevertheless, contrary to our findings, Spanish and Croatian studies found that females have greatly adapted their PA better during the lockdown than males [2,41].

### 4.2. Psychological Well-Being and Lockdown

Although isolation, social distancing, and other restrictive measures adopted by governments were effective in controlling the spread of the virus, they contributed to emotional and psychological disorders in different populations, which deserve to be addressed [42]. Thus, another aim of this study was to examine the impact of lockdown on the psychological well-being of the community university, paying special attention to differences in sex, role, PA levels, and age category. For this purpose, the PGWBI score was used as an overall score to determine the participants’ state of psychological well-being. In addition, each dimension of the psychological test was analyzed independently.

The COVID-pandemic by itself produced acute panic, anxiety, obsessive behaviors, depression, insomnia, and poor sleep quality [10,43]. Overall, our study found that university students were more affected than university employees, with females being severely affected over males in student groups, although no differences by gender in university employees were found.

Regarding university students, results in Figure 3 revealed a lower scoring of the five domains investigated (anxiety, vitality, self-control, positive well-being, and general health) in females as compared to male participants, which resulted in an overall reduction in the PGWBI score (60.2 vs. 70.1, respectively) which was maintained when participants were classified according to PA level (only for moderate and high PA levels), and age category (only for the young and young adult categories) (Table 2 and Table 3, respectively). In this sense, our results indicated a state of distress in female students, while males were more likely to be in a no distress state.

Our results are consistent with previous findings that showed gender-related poor mental health in athletes [44], the general population [45], and university students [10], with females being more affected than males [39]. However, some body of evidence did not find gender-related differences in psychological suffering [8,46]. Thus, the results obtained could be explained, in part, by the fact that the amount of PA performed during lockdown was higher in males than females (Figure 2D). In this sense, recent evidence supports the pivotal role of exercise and PA in decreasing different psychological symptoms such as stress, depression, and anxiety [47,48,49]. In the same way, the psychological benefits related to the practices of physical activity have been well established by the International Society of Sport Psychology (ISSP), with reduced psychosocial symptoms and negative effects, as well as an increase in the subjects’ self-esteem and positive emotions found among the main benefits [50]. Another potential explanation could be that females are more open about reporting their distress as compared to male participants [51].

On the other hand, our results show that the score associated with the psychological suffering dimensions such as anxiety and depression were higher in the employee than student groups (Figure 3). However, the employee groups also exhibited a higher score in the vitality, self-control, positive well-being, and general health dimensions, which resulted in an overall increase in the PGWBI score in both male (82.6 vs. 70.1; *p* < 0.001) and female groups (79.6 vs. 60.2; *p* < 0.001) as compared to the student groups (Figure 3). These findings may indicate a higher level of resilience in employees as compared to the student groups since resilience has been described as the ability to adapt successfully in the face of stress and adversity situations, such as the COVID-19 pandemic [52]. In other words, a person may feel psychological distress and nevertheless be resilient [46].

In this sense, the psychological well-being of the student population has been a major concern for several years. Before the COVID-19 pandemic, a national French survey of ~19,000 university students showed that 8% of the participants declared having suicidal thoughts in the previous 12 months, and 37% affirmed having experienced an episode of depression [53]. In addition, lower levels of engagement on campus, low energy levels, poorer relationships with other classmates, and lower average grades have been described as causes of the alteration of mental health [54], and this could explain, in part, the results of our study.

Along with this, a large body of evidence has shown that the COVID-19 outbreak has exacerbated this problem, with university students experiencing poorer mental health during lockdown [55,56,57]. In this regard, Elme and co-workers reported that among university students, the vulnerability of family members to COVID-19 infection was a source of worry [58]. Similarly, the fact of having to adapt to a new learning environment has been a stressor, particularly when associated with increased workloads and inadequate internet connectivity [57].

Finally, has been established that being young can contribute to being vulnerable to psychological distress and lower resilience during the pandemic [59]. In this way, younger people were more vulnerable to stress, anxiety, and depression during the COVID-19 pandemic [59]. For instance, [60] found that students in China were at a greater risk of stress, anxiety, and depression in response to the COVID-19 outbreak than older adults.

Despite being less at risk for serious complications from COVID-19, young adults appear to be more vulnerable to worsened mental well-being during the lockdown. In this sense, [61] reported a higher prevalence of depression and anxiety symptoms in younger people as compared to older adults at the beginning of the initial outbreak.

However, regarding the age category, we failed to find differences between different sub-groups according to the role in the PGWBI-S score or any dimension (intra-group category) (Table 2). A possible explanation is that our sample was not sufficiently heterogeneous in terms of age for being able to find the differences reported in the literature, as the average age of both the student group and the employee group was 23.8 and 39.7 years old, respectively.

## 5. Limitations

The study had several limitations that should be discussed and considered for future research. First, as the study was cross-sectional and a self-reported survey was used, some weaknesses could have been introduced because it may preclude the detection of possible bias in the measures of physical activity (before vs. during lockdown). Although the volume of physical exercise of each participant in the study was obtained from a standardized questionnaire, in the absence of physiological measurements of fitness, we cannot discard some inaccuracies in the time or the intensity of the exercise performed. Secondly, the sample of employee volunteers was limited and may have introduced bias. Furthermore, findings from this academic community may not necessarily apply to the general public or other academic communities who may be of lower socioeconomic, physical, or educational backgrounds.

## 6. Practical Applications

The fundamental application of our results focuses on the importance of implementing strategies to promote the practice of physical activity and improve psychological support within a university academic community during situations of stress and adversity, such as a lockdown. According to our results, in future situations similar to the COVID-19 pandemic, both government policies and the strategies of the university welfare teams should be both role- and gender-specific.

## 7. Conclusions

The main purpose of our study was to provide insight into the Colombian academic community’s PA levels and mental well-being during the COVID-19 lockdown. Both students and employees perceived that the COVID-19 lockdown negatively affected their total PA levels. Additionally, our results suggest that both sex and PA levels influenced psychological well-being during the lockdown, specifically with students being more affected than employees. Students, especially those who were female, may have a maladjusted psychological profile to confinement situations.

Furthermore, as this is the first study on the impact of the COVID-19 lockdown on PA levels and mental well-being in a Colombian academic community, these data could also be used as the baseline in order to compare them with other populations or for the design of PA and psychological recommendations during a prolonged at-home lockdown.

## Figures and Tables

**Figure 1 ijerph-19-11234-f001:**
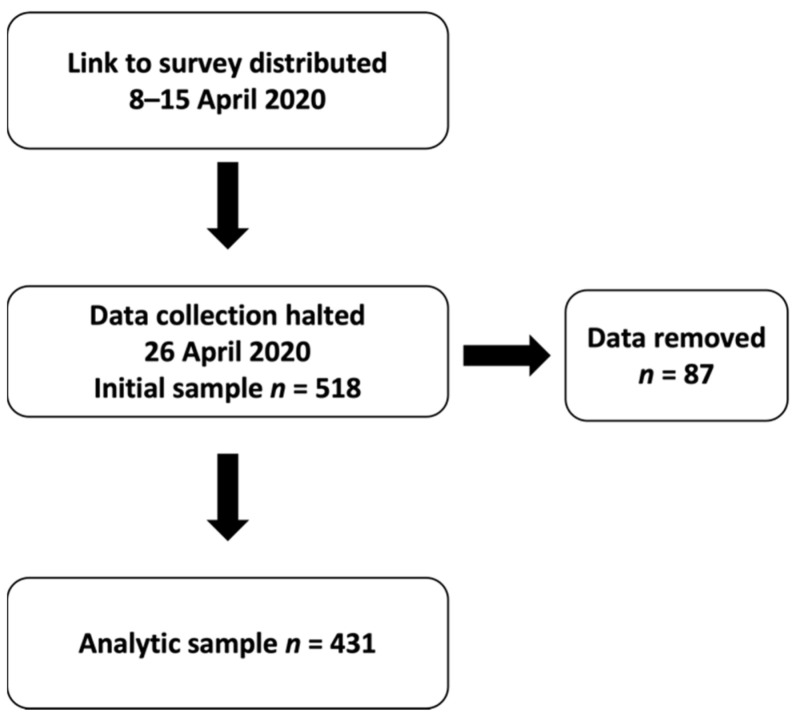
Flow diagram illustrating the timeline of participant recruitment and the removal of cases that had missing data.

**Figure 2 ijerph-19-11234-f002:**
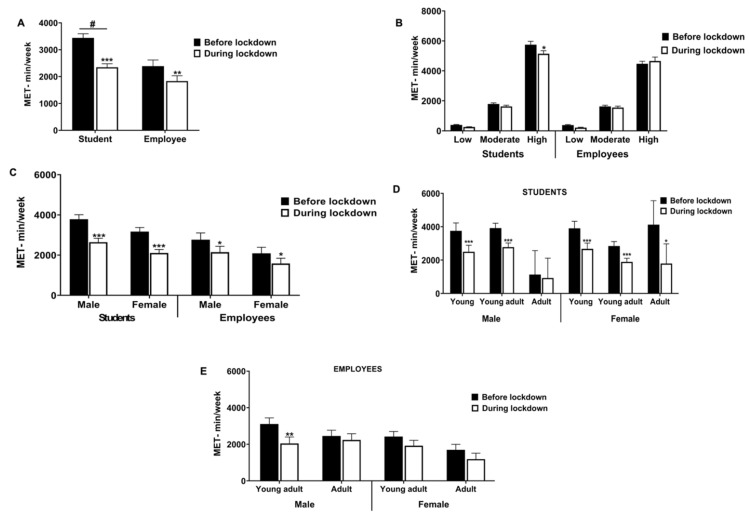
Changes in total physical activity in MET- minutes/week before and during lockdown. (**A**) Effects of lockdown in entire sample according to role. (**B**) Effects of lockdown in entire sample according to PA levels (**C**) Effects of lockdown in male and female participants, according to role. (**D**) Effects of lockdown is student groups according to age category. (**E**) Effects of lockdown in employee groups according to age category. Bars represent mean ± SEM. */**/*** denotes *p*  <  0.05/0.01/0.001, compared to before lockdown; # *p* < 0.05 compared to employee group.

**Figure 3 ijerph-19-11234-f003:**
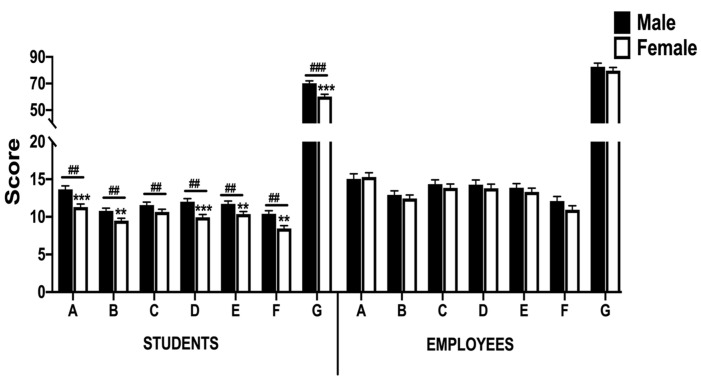
Score in the PGWBI-S test (PGWBI score and all six dimensions) in student and employee groups during lockdown. Bars represent mean ± SEM. A: anxiety; B: vitality; C: depression; D: self-control; E: well-being; F: general health; G: PGWBI score. A high PGWBI score represents greater well-being. **/*** denotes *p*  <  0.01/0.001 compared to male students; ##/### denotes *p*  <  0.01/0.001 compared to the employee group.

**Table 1 ijerph-19-11234-t001:** Characteristics of the participants (n = 431).

Categorical Variable	Students n (%)	Employees n (%)
Male	133 (44.6)	59 (44.4)
Female	165 (55.4)	74 (55.6)
Young	83 (27.9)	-
Young-Adult	207 (69.5)	68 (51.1)
Adult	8 (2.7)	65 (48.9)
Underweight	20 (6.7)	1 (8)
Normal	193 (64.8)	79 (59.4)
Overweight	70 (23.5)	41 (30.8)
Obese	15 (5)	12 (9)
Total	298 (69.1)	133 (30.9)

**Table 2 ijerph-19-11234-t002:** Mean of PGWBI score and dimensions for male and female according to age category.

Dimension	Group	Students			Employees	
		Young	Young Adult	Adult	Young Adult	Adult
Anxiety	Male	13.6 ± 0.8	13.3 ± 0.5 ^%%^	17.3 ± 2.6 ^%^	14.9 ± 1.0	14.8 ± 0.9
	Female	11.6 ± 0.7	11.2 ± 0.4	9.15 ± 2.6	15.4 ± 0.8 **	14.8 ± 0.9 ^#^
Vitality	Male	11.0 ± 0.7	10.5 ± 0.4 ^%^	10.0 ± 2.1	12.9 ± 0.8 ^$^	12.8 ± 0.7
	Female	10.1 ± 0.6	9.0 ± 0.4	10.0 ± 2.1	12.6 ± 0.6 **	11.7 ± 0.7
Depression	Male	12.3 ± 0.7	11.25 ± 0.4	9.1 ± 2.3	13.9 ± 0.8 ^$$^	14.4 ± 0.8 ^&^
	Female	11.1 ± 0.6	10.2 ± 0.4	13.7 ± 2.3	13.4 ± 0.7 **	13.9 ± 0.8
Self-control	Male	12.9 ± 0.8 ^%%^	11.6 ± 0.5 ^%^	10.0 ± 2.4	13.4 ± 0.9	14.7 ± 0.8
	Female	9.15 ± 0.72	9.9 ± 0.4	14.6 ± 2.4	13.8 ± 0.7 **	13.4 ± 0.8
Positive well-Being	Male	12.7 ± 0.7 ^%^	11.0 ± 0.4	14.6 ± 2.1	13.5 ± 0.8 ^$$^	14.0 ± 0.7
	Female	10.2 ± 0.6	10.1 ± 0.4	15.5 ± 2.2	13.1 ± 0.6 **	13.0 ± 0.7
General Health	Male	10.7 ± 0.8 ^%^	9.9 ± 0.5 ^%%^	12.8 ± 2.4	9.9 ± 0.5 ^$^	11.9 ± 0.8
	Female	8.6 ± 0.7	8.1 ± 0.4	10.9 ± 2.4	10.3 ± 0.7 *	11.1 ± 0.8
PGWBI score	Male	73.6 ± 3.6 ^%%^ (ND)	67.8 ± 2.2 ^%%^ (D)	74.1 ± 0.9 (ND)	81.1 ± 4.1 ^$$^ (ND)	82.8 ± 3.9 (ND)
	Female	61.0± 3.2 (D)	58.9 ± 2.0 (SD)	74.1 ± 10.9 (ND)	78.8 ± 3.4 ** (ND)	78.2 ± 3.7 (ND)

Abbreviations: SD: Strong Distress; D: Distress; ND: No Distress; %/%% denotes *p* < 0.05/0.01 vs. female student; */**/# denotes. *p* < 0.05/0.01/0.05 vs. respective young adult/adult female student groups; $/$$/& denotes *p* < 0.05/0.01/0.05 vs. young adult/adult.

**Table 3 ijerph-19-11234-t003:** Mean of PGWBI score and dimensions for male and female according to PA level during Lockdown.

Dimension	Group	Students			Employees		
		Low	Moderate	High	Low	Moderate	High
Anxiety	Male	11.9 ± 0.8 *	13.7 ± 0.7 ^%%^	14.7 ± 0.7	13.2 ± 1.2	15.6 ± 1.0	15.2 ± 1.2
	Female	9.9 ± 0.7 ^##^	10.7 ± 0.6 ^$$^	13.7 ± 0.7	13.6 ± 0.9 ^††^	15.7 ± 0.9 ^††^	17.2 ± 1.3 ^†^
Vitality	Male	8.4 ± 0.6 **	10.1 ± 0.6 ^††%^	12.8 ± 0.5	13.2 ± 1.0 ^&&^	12.1 ± 0.8 ^&^	13.6 ± 0.6
	Female	8.4 ± 0.5 ^##^	8.5 ± 0.5 ^$$^	11.6 ± 0.5	10.8 ± 0.7 ^⁋⁋††^	12.3 ± 0.7 ^††^	14.9 ± 1.0 ^††^
Depression	Male	9.9 ± 0.7 **	10.6 ± 0.6 ^††^	13.4 ± 0.6	14.4 ± 1.1 ^&&^	14.2 ± 0.9 ^&&^	14.0 ± 1.0
	Female	9.9 ± 0.6	10.0 ± 0.5	11.9 ± 0.6	13.2 ± 0.8 ^††^	13.6 ± 0.8 ^††^	14.6 ± 1.2
Self-control	Male	11.2 ± 0.7	11.7 ± 0.7 ^%^	12.8 ± 0.6	13.9 ± 1.2	13.7 ± 0.9	14.8 ± 1.1
	Female	9.39 ± 0.6	9.1 ± 0.6	11.3 ± 0.7	13.0 ± 0.8 ^††^	14.0 ± 0.8 ^††^	14.1 ± 1.3
Positive well-Being	Male	9.6 ± 0.6 **	11.4 ± 0.6 ^%^	13.3 ± 0.6	13.4 ± 1.0 ^&&^	13.9 ± 0.8 ^&^	14.0 ± 1.0
	Female	10.0 ± 0.5	9.2 ± 0.5 ^$$^	12.0 ± 0.6	11.9 ± 0.7	13.7 ± 0.7 ^††^	14.1 ± 1.1
General Health	Male	9.1 ± 0.7 **	8.9 ± 0.7 ^††^	12.3 ± 0.6 ^%^	12.5 ± 1.1 ^&^	11.2 ± 0.9	12.8 ± 1.1
	Female	8.4 ± 0.6	7.3 ± 0.5 ^$$^	9.8 ± 0.6	10.0 ± 0.8	10.8 ± 0.8 ^††^	12.0 ± 1.2
PGWBI score	Male	60.3 ± 3.3 **	66.6 ± 3.1 ^††%%^	79.5 ± 2.9 ^%^	80.9 ± 5.2 ^&&^	80.9 ± 4.2 ^&&^	84.5 ± 4.9
	Female	56.2 ± 2.8 ^##^	55.1 ± 2.6 ^$$^	70.5 ± 3.0	72.8 ± 3.8 ^††^	80.3 ± 3.8 ^††^	87.0 ± 5.6 ^††^

Abbreviations: SD: Strong Distress; D: Distress; ND: No Distress; %/%% denotes *p* < 0.05/0.01 vs. female student; */**/†† denotes *p* < 0.05/0.01 vs. high male students, respectively; $$/## denotes *p* < 0.01/0.01/ vs. high female students, respectively; ⁋⁋ denotes *p* < 0.01 vs. high female employees; †/†† denotes *p* < 0.05/ 0.01 vs. respective student female group; &/&& denotes *p* < 0.05/0.01 vs. respective student male group.

## Data Availability

All the data are presented in this study.

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
