# Peer review of "Physical Activity Levels and Psychological Well-Being during COVID-19 Lockdown among University Students and Employees"

_ijerph, 2022, doi:10.3390/ijerph191811234_

Round 1
Reviewer 1 Report
Thank you for the opportunity to review the manuscript entitled “Physical activity levels and psychological well-being during COVID-19 lockdown among university students and employees.” This manuscript details a cross-sectional analysis that describes physical activity levels and psychological well-being, stratified by working status, and sex in a sample of Colombian participants. It is unclear what this study adds to the literature and current state of knowledge/ it is unclear what the novelty of this work is. Furthermore, it is unclear if authors adjusted analyses for covariates. This information does not seem to be provided in the methods or results, making it difficult to interpret findings and assess manuscript quality. Please see below for specific comments:
Abstract:
1. Please state the study design in the abstract for the reader. Was it a cross-sectional analysis? A natural experiment?
Introduction:
2. Based on the introduction, it is not clear what this study adds, or what gap in the literature this study is trying to fill. It is well-documented that the pandemic had adverse effects on physical activity levels and psychological outcomes. Please frame the introduction so that it sets up a nice argument for what this study adds. Perhaps this is one of the first studies among a Columbian sample? Perhaps this is the first study to directly compare students to employees? I’m not sure.
Methods:
3. Do you have any more detailed information on the study participant flow, to give the reader a better sense of how you arrived at your analytic sample of 431 participants? For example, how many potential participants were contacted, how many agreed/declined to participate, did any people agree to participate but were ineligible, etc.? Please report this in the manuscript, as right now you only report those with incomplete vs. complete data.
4. Please provide a reference or rationale for your age categories, otherwise, they seem rather arbitrary.
5. Please place the procedures subsection above where you describe your measures.
6. Please move the internal consistency analysis for the PGWB-S to the same section where the measure is initially described.
Results:
7. Figure 2: Do panels A and C reflect the same information? I would suggest removing panel C since it doesn’t add information and it is the only panel that is in a different format than the rest.
8. Do not interpret your results as subheadings. For example, “3.2. Psychological well-being is strongly influenced by role and sex” should just read as “Factors that relate to psychological well-being.” Additionally, avoid using the term “moderate” (subheading 3.1.) unless statistical moderation was conducted. “Relate to” is a better term to use here.
9. Figure 3: Please remind the reader whether high scores represent greater well-being or less well-being.
10. Were models adjusted for covariates? It seems like there are a lot of covariates missing from this analysis, such as weight status, socioeconomic status, etc. It is unclear whether this information was accidentally left out of the statistical analysis section of the manuscript or whether models really did not adjust for these factors. This makes it difficult to assess the quality of this manuscript.
General:
Please make sure the manuscript is professionally edited by a native English speaker.
Author Response
We would like to thank the reviewer for the critical evaluation and valuable comments. We have considered the recommendations and suggestions. Itemized responses are listed below. All the modifications have been marked up throughout the manuscript using the track changes function as the editor recommended.
- Please state the study design in the abstract for the reader. Was it a cross-sectional analysis? A natural experiment
Answer: The following lines (27 -30) have been added to the abstract:
A cross sectional study was conducted through an online survey during the first isolation. 431 respondents (192 males) aged 18–60 years old (28.1± 11.1 yrs) participated. The international Physical Activity Questionnaire (IPAQ), and the short version of Psychological General Well-Being Index (PGWBI-S) were used.
- Based on the introduction, it is not clear what this study adds, or what gap in the literature this study is trying to fill. It is well-documented that the pandemic had adverse effects on physical activity levels and psychological outcomes. Please frame the introduction so that it sets up a nice argument for what this study adds. Perhaps this is one of the first studies among a Columbian sample? Perhaps this is the first study to directly compare students to employees? I’m not sure.
Answer: Based of the reviewer’s comment, the following text was added on lines 92 to 94
Despite the aforementioned, no studies have reported the impact of COVID-19 lockdown on PA levels and psychological well-being in a sample from a Colombian University’s Academic Community.
- Do you have any more detailed information on the study participant flow, to give the reader a better sense of how you arrived at your analytic sample of 431 participants? For example, how many potential participants were contacted, how many agreed/declined to participate, did any people agree to participate but were ineligible, etc.? Please report this in the manuscript, as right now you only report those with incomplete vs. complete data.
Answer: The following information has been added to lines 121 to 124 in the new version of the manuscript.
Survey data collection was halted on 26/04/20, and the data was downloaded. Any participant that was missing data for one or more of the items of interest (i.e., physical activity, sedentary behavior, psychological well-being, age, bodyweight, height, etc.) in the survey was subsequently removed from the data set.
Additionally, figure 1 was modified by adding one box related to the data removed and moved to the results section.
Finally, regarding to the study participant flow, we have added the following text on the results section, to lines 209 to 211:
The potential participants were n = 518 (Figure 1). However, after excluding 87 datasets because of missing values for physical activity, psychological well-being, age, bodyweight, height or sex, the final sample was n= 431
- Please provide a reference or rationale for your age categories, otherwise, they seem rather arbitrary.
Answer: we have included reference 8 on line 140 from “Maugeri, G.; Castrogiovanni, P.; Battaglia, G.; Pippi, R.; D’Agata, V.; Palma, A.; Di Rosa, M.; Musumeci, G. The Impact of Physical Activity on Psychological Health during Covid-19 Pandemic in Italy. Heliyon 2020, 6, e04315, doi:10.1016/j.heliyon.2020.e04315.”
- Please place the procedures subsection above where you describe your measures.
Answer: Done
- Please move the internal consistency analysis for the PGWB-S to the same section where the measure is initially described.
Answer: Done
- Figure 2: Do panels A and C reflect the same information? I would suggest removing panel C since it doesn’t add information and it is the only panel that is in a different format than the rest.
Answer: Done
- Do not interpret your results as subheadings. For example, “3.2. Psychological well-being is strongly influenced by role and sex” should just read as “Factors that relate to psychological well-being.” Additionally, avoid using the term “moderate” (subheading 3.1.) unless statistical moderation was conducted. “Relate to” is a better term to use here.
Answer: We thank the reviewer for this comment, but we believe that writing the title in this manner facilitates the interpretation of the results by any reader. On the other hand, the term “moderate” refers to physical activity levels rather to any statistical aspect in this context.
- Figure 3: Please remind the reader whether high scores represent greater well-being or less well-being.
Answer: Done. Please look at line 275.
- Were models adjusted for covariates? It seems like there are a lot of covariates missing from this analysis, such as weight status, socioeconomic status, etc. It is unclear whether this information was accidentally left out of the statistical analysis section of the manuscript or whether models really did not adjust for these factors. This makes it difficult to assess the quality of this manuscript.
Answer: Previously, we performed the MANOVA test according to the weight and body mass index (BMI) variables, but no statistical differences were found. On account of this we have added the following text to line 279:
For weight and BMI covariates, no additional changes were found (data not shown).
General: Please make sure the manuscript is professionally edited by a native English speaker.
Answer: The manuscript has also been edited by a professional editor
Reviewer 2 Report
Dear Authors,
The manuscript is intriguing, but I recommend following guidelines for a higher quality display such as CROSS.
https://www.researchgate.net/publication/353522424_Checklist_for_Reporting_Of_Survey_Studies_CROSS
In the abstract, describe the study population and Describe all questionnaire instruments.
27 what kind of isolation? There have been several lockdowns, at different times .. enter the dates
30 MET?
35 I suggest softening the conclusions
Introduction
45 missing references..
47 Reference 2 makes no mention of the effectiveness of the lockdown
52 I suggest the adjunct with refs “On the other hand, the lockdowns have impacted all areas of society from education, sports, and community life.” ( refs: https://doi.org/10.3390/publications8040048 ; https://doi.org/10.23736/s0022-4707.21.12903-2 ; http://dx.doi.org/10.1136/bjsports-2020-102575 )
Methods
93 Specify the study design in the methods section with a commonly used term (e.g., cross-sectional or longitudinal).
all the participants section is not appropriate for the methods, they are the results of a selection. I recommend: Describe the study population (i.e., background, locations, eligibility criteria for participant inclusion in survey, exclusion criteria).
Provide information of survey’s time frame, such as periods of recruitment, exposure, and follow-up days. Describe the sampling techniques used (e.g., single stage or multistage sampling, simple random sampling, stratified sampling, cluster sampling, convenience sampling). Specify the locations of sample participants whenever clustered sampling was applied. Figure 1 is appropriate but if the results are exposed then move it into the results
Discussion and Conclusions
Interpretations: Give a cautious overall interpretation of results, based on potential biases and imprecisions and suggest areas for future research.
Generalizability : Discuss the external validity of the results.
Author Response
We would like to thank the reviewer for the critical evaluation and valuable comments. We have considered the recommendations and suggestions. Itemized responses are listed below. All the modifications have been marked up throughout the manuscript using the track changes function as the editor recommended.
- The manuscript is intriguing, but I recommend following guidelines for a higher quality display such as CROSS.
https://www.researchgate.net/publication/353522424_Checklist_for_Reporting_Of_Survey_Studies_CROSS
Answer: Following the reviewer's suggestion, we have checked the guidelines from the Checklist for Reporting Of Survey Studies (CROSS) document, and the following information has been added to lines 102 to 104 in the new version of the manuscript:
This was an observational and analytical cross‐sectional research study, where an electronic survey was sent to the isolated academic community via email for completion. A non-probability sampling method was used.
- In the abstract, describe the study population and Describe all questionnaire instruments
Answer: Done. Please take a look at lines 28 to 30 in the new version of the manuscript.
- 30 MET
Answer: That is right. The MET refers to the amount of energy expended when carrying out physical activity. According to the guidelines of International Physical Activity Questionnaire (IPAQ) scoring protocol, a continuous score is suggested to be expressed as: MET-min per week[1]. We have clarified it in the text.
- 35 I suggest softening the conclusion
Answer: We thank the reviewer for this comment, but we believe that these two lines are already soft.
- 45 missing reference
Answer: We have added reference number 2 in the new version of the manuscript: Rodríguez-Larrad, A.; Mañas, A.; Labayen, I.; González-Gross, M.; Espin, A.; Aznar, S.; Serrano-Sánchez, J.A.; Vera-Garcia, F.J.; González-Lamuño, D.; Ara, I.; et al. Impact of COVID-19 Confinement on Physical Activity and Sedentary Behaviour Inspanish University Students: Ole of Gender. International Journal of Environmental Research and Public Health 2021, 18, 1–14, doi:10.3390/ijerph18020369.
- 47 Reference 2 makes no mention of the effectiveness of the lockdow
Answer: We have changed that reference for this one: Ji, T.; Chen, H.L.; Xu, J.; Wu, L.N.; Li, J.J.; Chen, K.; Qin, G. Lockdown Contained the Spread of 2019 Novel Coronavirus Disease in Huangshi City, China: Early Epidemiological Findings. Clinical Infectious Diseases: An Official Publication of the Infectious Diseases Society of America 2020, 71, 1454–1460, doi:10.1093/CID/CIAA390.
This is reference number 3 in the new version of the manuscript
- 52 I suggest the adjunct with refs “On the other hand, the lockdowns have impacted all areas of society from education, sports, and community life.” ( refs: https://doi.org/10.3390/publications8040048; https://doi.org/10.23736/s0022-4707.21.12903-2 ; http://dx.doi.org/10.1136/bjsports-2020-102575
Answer: Done
- 93 Specify the study design in the methods section with a commonly used term (e.g., cross-sectional or longitudinal). all the participants section is not appropriate for the methods, they are the results of a selection. I recommend: Describe the study population (i.e., background, locations and inclusion criteria eligibility criteria for participant inclusion in survey, exclusion criteria).
Provide information of survey’s time frame, such as periods of recruitment, exposure, and follow-up days. Describe the sampling techniques used (e.g., single stage or multistage sampling, simple random sampling, stratified sampling, cluster sampling, convenience sampling). Specify the locations of sample participants whenever clustered sampling was applied. Figure 1 is appropriate but if the results are exposed then move it into the results.
Answer: Reviewer 2 is addressing two different issues in this query: methods and results sections. We have decided to answer them separately.
First, in the new version of the manuscript, we have added the following information to lines 102 to 104:
This was an observational and analytical cross‐sectional research study, where an electronic survey was sent to the isolated academic community via email for completion. A non-probability sampling method was used.
In addition, in the sub-section “2.2 procedures” we have described the study population including their background, location, inclusion criteria, sampling technique, and detailed the surveys’ time frame on lines 107 to 142.
Secondly, with respect to the figure 1 issue, we have moved it to the results section.
The following paragraph, including table 1, was moved to the final part of the procedures section:
Participants were divided into three age groups: young <21 (n =83); young adults aged 21–40 years old (n = 275), and adults aged 41–60 years old (n =75) [8]. In the employee groups, no young adult participants were included, as they were not part of the university's employee population (Table 1).
Additionally, we have re-written the following text in the results section, on lines 207 to 212: The potential participants were n = 518 (Figure 1). However, after excluding 87 datasets because of missing values for physical activity, psychological well-being, age, body weight, height or sex, the final sample was n= 431 (students n= 298 [69.1%] and employees n= 133 [30.9%], mean age 23.7 ± 6.5 years old and 39.9 ± 11.0 years old; mean height 168.1 ± 25.2 cm and 166.3 ± 8.2 cm; mean weight 64.7 ± 12.9 kg and 69.4 ± 12.9 kg; and BMI 23.1 ± 3.7 and 25.0 ± 3.7, respectively), which underwent detailed analysis.
- Interpretations: Give a cautious overall interpretation of results, based on potential biases and imprecisions and suggest areas for future research
Answer: We thank the reviewer for this comment. In the manuscript, this issue was already answered in section 5 (limitations). Moreover, we have added the following text on line 479, in section 7 (conclusions):
… the baseline in order to compare them with other populations, or for the …
- Generalizability: Discuss the external validity of the results
Answer: We thank the reviewer for this comment. We had already made a comment on lines 458 to 461 in the limitations section.
Round 2
Reviewer 1 Report
Thank you for addressing my comments and suggestions. The manuscript is much improved. One final minor suggestion:
The title of the last oval of the flow diagram is misleading. Please rename this to "analytic sample."
Author Response
Thank you for addressing my comments and suggestions. The manuscript is much improved. One final minor suggestion:
Thank you for all your constructive and valuable feedbacks. We truly believe that after considering all your concerns, the quality of the manuscript has been significantly improved.
The title of the last oval of the flow diagram is misleading. Please rename this to "analytic sample."
Thank you for your suggestion. We have renamed the title of the last oval of the flow diagram to "analytic sample".
Reviewer 2 Report
Dear authors, thank you for the careful and detailed review of my concerns
Author Response
Dear authors, thank you for the careful and detailed review of my concerns
Thank you for all your constructive and valuable feedbacks. We truly believe that after considering all your concerns, the quality of the manuscript has been significantly improved.